# Impact of the COVID-19 Pandemic on Prostate Cancer Diagnosis, Staging, and Treatment: A Population-Based Study in Northern Italy

**DOI:** 10.3390/biology13070499

**Published:** 2024-07-04

**Authors:** Lucia Mangone, Francesco Marinelli, Isabella Bisceglia, Angelina Filice, Maria Barbara Braghiroli, Francesca Roncaglia, Andrea Palicelli, Fortunato Morabito, Antonino Neri, Roberto Sabbatini, Cinzia Iotti, Carmine Pinto

**Affiliations:** 1Epidemiology Unit, Azienda Unità Sanitaria Locale—IRCCS di Reggio Emilia, 42122 Reggio Emilia, Italy; francesco.marinelli@ausl.re.it (F.M.); isabella.bisceglia@ausl.re.it (I.B.); mariabarbara.braghiroli@ausl.re.it (M.B.B.); francesca.roncaglia@ausl.re.it (F.R.); 2Nuclear Medicine Unit, Azienda Unità Sanitaria Locale—IRCCS di Reggio Emilia, 42122 Reggio Emilia, Italy; angelina.filice@ausl.re.it; 3Pathology Unit, Azienda Unità Sanitaria Locale—IRCCS di Reggio Emilia, 42122 Reggio Emilia, Italy; andrea.palicelli@ausl.re.it; 4Gruppo Amici Dell’Ematologia Foundation—GrADE, 42123 Reggio Emilia, Italy; fortunato.morabito@grade.it; 5Scientific Directorate, Azienda Unità Sanitaria Locale—IRCCS di Reggio Emilia, 42122 Reggio Emilia, Italy; antonino.neri@ausl.re.it; 6Medical Oncology Division, Department of Oncology and Hematology, University of Modena and Reggio Emilia, 41125 Modena, Italy; sabbrob@unimore.it; 7Radiation Oncology Unit, Department of Advanced Technology, Azienda Unità Sanitaria Locale—IRCCS di Reggio Emilia, 42122 Reggio Emilia, Italy; cinzia.iotti@ausl.re.it; 8Medical Oncology Unit, Azienda Unità Sanitaria Locale—IRCCS di Reggio Emilia, 42122 Reggio Emilia, Italy; carmine.pinto@ausl.re.it

**Keywords:** prostate cancer, COVID-19, stage, Gleason, surgery, radiotherapy

## Abstract

**Simple Summary:**

COVID-19 has had a dramatic impact on new cancer diagnoses and the treatment of cancer patients. This study examines how the COVID-19 pandemic affected prostate cancer diagnosis, staging, and treatment in Reggio Emilia, Northern Italy. Using data from the Cancer Registry from 2018 to 2021, we found a significant drop in new prostate cancer diagnoses during 2020, the height of the pandemic. The number of new cases decreased by 31% in 2020, with a slight recovery of 5% in 2021. We also observed a shift towards more advanced stages and aggressive forms of prostate cancer, with fewer early-stage diagnoses and more cases of metastatic cancer. Despite these changes, treatments such as surgery and radiation therapy have remained constant. The pandemic disrupted an increasing trend of prostate cancer diagnoses seen before 2019, highlighting the need for ongoing diagnostic services and healthcare delivery, even during global health emergencies. Our study highlights the significant impact of COVID-19 on prostate cancer management and the importance of healthcare system resilience.

**Abstract:**

The COVID-19 pandemic has caused delays in cancer diagnoses and reductions in treatments. The aim of this work is to evaluate the impact of the pandemic on prostate cancer by evaluating whether there has been a shift towards more aggressive (Gleason) and more advanced tumors (stage IV) and a decline in treatments. The study was conducted on 1123 cases of prostate cancer incident in the Province of Reggio Emilia, Northern Italy, in the period of 2018–2021. In 2020, there was a decline in new diagnoses of prostate cancer (−31%), followed by a slight recovery in 2021 (+5%). While Gleason 7 and 8–10 values remained constant, a significant decrease was recorded in stage I (38.7%, 41.6%, 35.5%, and 27.7%) and an increase in stage IV (13.1%, 13%, 15.4%, and 20%) cases in the years 2018, 2019, 2020, and 2021, respectively. However, there was no impact on surgical treatment (which remained constant at around 35%) and radiotherapy (around 39%). Our findings underline the profound impact of COVID-19 on prostate cancer management, highlighting the importance of healthcare resilience in the face of unprecedented disruptions.

## 1. Introduction

Prostate cancer represents the most prevalent neoplasm in men in the world [1]. In Italy, it remains the leading neoplasm among men, with approximately 36,000 new diagnoses annually, accounting for 19% of all cases [2]. Mortality from prostate cancer in Italy amounts to about 7700 deaths every year, constituting 8% of total cancer-related deaths. The 5-year survival rate is notably high at around 92%, with no significant variation across different age groups and geographical areas [3]. The COVID-19 pandemic, which emerged in 2020, substantially impacted the management of oncological diseases, with this being also related to its rapidly evolving viral replication and its transmission between individuals [4]. This impact manifested as interruptions in cancer treatment, prevention, and monitoring activities as well as the repurposing of many cancer facilities to COVID-19 units, leading to delays in both diagnosis and treatment [5,6]. An initial analysis by Liang [7] highlighted the effects of SARS-CoV-2 infections on cancer patients in China, revealing higher Intensive Care Unit admissions and mortality rates among cancer patients, especially those with recent cancer diagnoses. Subsequent studies reinforced these findings, highlighting delays in diagnosis and treatments and their consequent impacts on cancer incidence and mortality across different countries and cancer types [8,9,10,11,12,13]. During the pandemic, cancer patients exhibited an increased risk of mortality, especially among the elderly, males, individuals with comorbidities, smokers, and people with a low-performance status [14]. In the UK, patients with cancer undergoing chemotherapy experienced a 1.5-times higher risk of mortality, even if not statistically significant [15]. Italian studies corroborated these observations, indicating an increased risk of hospitalization and mortality from COVID-19 among cancer patients compared to the general population, with pronounced risks for lung, breast, and hematological cancers [16]. Concerning cancer incidence, one Italian study [17] reported a 45% reduction in new cancer diagnoses during the lockdown period (March–May 2020) compared to the same months in 2018–2019, with a significant decrease in diagnoses of skin cancers and melanomas (−57%), colorectal (−47%), prostate (−45%), and lung (−27%) cancers. A subsequent study evaluated the impact of the lockdown and the suspension of screening tests on new cancer diagnoses, highlighting a 35% decrease in new diagnoses, particularly breast (−35%), prostate (−32%), and lung (−22%) cancers [18].

A decline in the incidence of prostate cancer has also been reported in Brazil (−27.3%9 [19] and in the Netherlands Cancer Registry (−17%) [20], which confirms also that the stability of the stage at diagnosis and radical prostatectomies performed in 2020 was comparable to that in 2018–2019.

The current study aims to analyze the influence of the COVID-19 pandemic on new prostate cancer diagnoses, focusing on changes in Gleason score, stage, and treatments.

## 2. Materials and Methods

### 2.1. Setting and Data Sources

This study utilizes data from the Cancer Registry (CR) of the province of Reggio Emilia, located in Northern Italy, which serves a population of 532,000 inhabitants. The CR is distinguished by its high-quality data, evidenced by its high percentage of microscopic confirmations (93.3%) and the absence of death-certificate-only (DCO) data [21]. Additionally, the registry contains recent data, updated to 2021. The activities of the CR have been sanctioned by the provincial Ethics Committee of Reggio Emilia (protocol no. 2014/0019740 of 4 August 2014). The primary sources of data for the CR include anatomical pathology reports, hospital discharge records, and mortality data, which are integrated with laboratory tests, diagnostic reports, and information from general practitioners.

### 2.2. Data Collection

This study encompasses all prostate cancers recorded in the CR from 2018 to 2021. Prostate cancer cases were identified based on the International Classification of Diseases for Oncology, Third Edition (ICD-O-3), with the topography C61 [22]; most cases had an adenocarcinoma morphology (ICD-O-3: 8140/3). In addition to the variables of tumor site and morphology routinely collected by Cancer Registries (CRs), extensive efforts were undertaken to obtain detailed information on the stage, Gleason score, and treatment of each case by thoroughly reviewing medical records. Specifically, data concerning the stage (TNM 8th Edition) [23], Gleason score (6 = low grade; 7 = intermediate grade; 8–10 = high grade), surgical interventions, and radiotherapy were meticulously extracted from hospital medical records. To provide a comprehensive overview of prostate cancer trends, this study extended beyond the recent four-year period (2018, 2019, 2020, 2021) to include incidence and mortality trends over the past ten years. Descriptive analyses of patient characteristics, including their age at diagnosis, divided into three groups: ≤70, 71–80, and 80+; Gleason score (low grade, intermediate grade, high grade, or unknown); stage (I, II, III, IV, or unknown); surgery (yes, no); and radiotherapy (yes, no), were conducted according to the year of diagnosis. We also present descriptive analyses for the variables reported above, stratified for the four years of diagnoses.

### 2.3. Statistical Analyses

To determine the difference between the patients’ characteristics and the years of diagnosis, we performed a chi-squared test for surgery and radiotherapy and a test for assessing trends in the age at diagnosis, Gleason score, and stage using Poisson regression models with the period as the independent continuous variable. The standardized incidence and mortality rates for the period 2010–2021 were calculated. Population estimates, used to derive these rates, were based on the general population of the Province of Reggio Emilia as of 1 January of each year. Incidence rates and incidence-based mortality rates were adjusted to the 2013 European standard population and expressed per 100,000 people each year. Results are reported with 95% confidence intervals (CIs), and a *p*-value < 0.05 was considered as statistically significant. Trends over time were analyzed by calculating the annual percent change (APC) in age-standardized rates using joinpoint regression [24]. The APC is one way to characterize the trends in cancer rates over time: the output includes the estimated annual percentage rate change. With this approach, the cancer rates are assumed to change at a constant percentage of the rate of the previous year. As for the intervals used, joinpoint fits the selected trend data (incidence and mortality rates) into the simplest joinpoint model that the data allow. It is not always reasonable to expect that a single APC can accurately characterize the trend over an entire series of data. The joinpoint model uses statistical criteria to determine when and how often the APC changes. The maximum number of joinpoints predicted for these analyses was fixed to four. Analyses were performed using Stata 16.1 SE (Stata Corp, College Station, TX, USA).

## 3. Results

Between 2018 and 2021, in the Province of Reggio Emilia, a total of 1123 cases of prostate cancer were registered. The distribution of these cases by age, Gleason score, stage, surgical intervention, and radiotherapy is detailed in Table 1. The mean age at diagnosis was 71.8 years (±sd 8.3). Moreover, 43.5% of cases were diagnosed between 71 and 80 years old, 42.1% were diagnosed aged ≤ 70 years old, and 14.4% were diagnosed aged 80+. The majority of tumors (51.9%) had a Gleason score of 7, 25.5% had a Gleason score of 6, and 21.9% had a Gleason score of 8–10. Only eight cases (0.7%) showed an unknown value. Concerning the stage distribution, 36.5%, 26.2%, 17.5%, and 15.1% were diagnosed in stages I, II, III, and IV, respectively. In 53 cases (4.7%), the stage was unknown. In 393 patients (35%), a radical prostatectomy was performed, and in 429 patients (38.2%), radiotherapy was performed.

Table 2 shows the annual distribution of cases. The age-wise distribution remained largely consistent, with notable shifts in the 71–80 and 80+ categories: 47.5% (2018), 44.8% (2019), 42.3% (2020), and 38.0% (2021) for the former, and 11.8% (2018), 16.5% (2019), 10.3% (2020), and 18.8% (2021) for the latter. Gleason scores remained relatively consistent over the years. However, in 2021, there was a decrease in the percentage of cases classified as Gleason 6 compared to that in 2020 (18.4% in 2021 and 28.2% in 2020). Conversely, there was an increase in the percentage of cases classified as Gleason 7, rising from 49.6% in 2020 to 54.7% in 2021, and in the percentage of cases classified as Gleason 8–10, which increased from 21.4% in 2020 to 25.7% in 2021. The distribution by stage shows a significant decrease in stage I tumors (38.7%, 41.6% to 35.5% and 27.7%; *p* < 0.05) and a significant increase in stage II (23.9%, 21.8%, 30.8%, and 30.6%; *p* < 0.05) and stage IV (13.1%, 13%, 15.4%, and 20%; *p* < 0.05) tumors in the years 2018–2021, respectively. Surgical intervention rates remained relatively stable at around 35% throughout the period, while radiotherapy usage increased significantly over the years, despite a slight decline in 2020.

To verify the impact that COVID-19 has had on patients undergoing radical prostatectomy, only operated patients were examined. As regards age, there was a decline in surgery in patients aged 71–80 in the years 2019–2021 (surgery = 40.5%, 36.5%, and 27.4%, respectively). Concerning stage, however, in 2020, there was an increase in patients operated on within the stage I and II groups, while for stage III, half the amount of patients were operated on compared to the previous year (44.8% in 2019 and 21.2% in 2020) (Table 3).

To contextualize the incidence and mortality trends of prostate cancer over the entire period, Figure 1 illustrates a significant decline in the incidence from 2010 to 2014 (APC −5.7; 95% CI -7.9; -3.5; *p* < 0.001), followed by a significant rise in incidence from 2014 to 2019 (APC 5.8; 95% CI: 3.5–8.3; *p* < 0.001). This trend was sharply disrupted during the COVID-19 lockdown, with the APC dropping to −13.9% (95% CI −17.1;–10.5; *p* < 0.001). Mortality rates remained stable over the entire period considered in this study (APC 1; 95% CI −2.3; –4.5; *p* = 0.52).

## 4. Discussion

The COVID-19 pandemic triggered widespread disruptions in healthcare systems globally, profoundly impacting the diagnosis and management of various diseases, including prostate cancer. Our study focused on assessing the repercussions of pandemic-induced lockdown measures on prostate cancer incidence and treatment outcomes. The province of Reggio Emilia, known for its high incidence of cancer and significant impact during the initial wave of the COVID-19 pandemic [25,26], provided a reasonable setting for our study. Utilizing real-world data, we aimed to evaluate the impact of the COVID-19 pandemic on new prostate cancer diagnoses, in terms of stage, Gleason score, and treatment approaches. Data from the Cancer Registry covering the years 2018–2019 (pre COVID-19 pandemic), 2020 (the peak pandemic year), and 2021 (post-pandemic recovery) were used. The COVID-19 pandemic has exerted a significant influence on the incidence of prostate cancer, resulting in a notable reduction of 31% during the pandemic year. This observed decline within our cohort is consistent with broader global trends. For instance, in the UK, during the pandemic in 2020, the incidence of prostate cancer dropped by 4772 cases (31%) and began to recover in 2021, reducing the gap to 18% (3148 cases), and by 2022, the incidence returned to the expected levels [27]. In the Netherlands, a 17% decline in prostate cancer diagnoses during the first COVID-19 wave was followed by a return to approximately 95% of the expected diagnoses by the end of 2020 [20]. Conversely, an Australian study found that most regions, except Victoria, were unaffected in terms of prostate cancer testing during 2020. Victoria experienced a statistically significant decrease in the number of PSA tests, correlating with the extended lockdown that occurred in this state [28]. In Sweden, a 36% reduction in prostate cancer cases was registered in 2020 compared with the corresponding period in previous years, with the decline being more pronounced in men over aged 75 years (−51%) than in men aged 70–75 (−37%) and men below the age of 70 (−28%) [29]. Our study showed no significant variations in the age at diagnosis across the pre- and post-pandemic years, with the average age remaining around 71 years. This finding contrasts with other studies, such as one from England, which reported a slight increase in the average age at diagnosis during the pandemic [27]. However, the reduction in the incidence of prostate cancer observed in our study among the 71–80 age group during the pandemic year somehow aligns with the results of the Lemanska study [27].

Additional notable effects observed in our study were the shifts in higher-risk Gleason scores and more advanced stages during the COVID-19 pandemic. Specifically, there was a rise in Gleason 7 cases from 53.1% in 2018 to 54.7% in 2021, and in Gleason 8 cases from 20.3% in 2018 to 25.7% in 2021. Similarly, before the pandemic, a substantial proportion of prostate cancer cases were diagnosed at an early stage, reflecting the effectiveness of early detection efforts. In 2018, for example, 38.7% of cases were identified at stage I; however, during the pandemic, the percentage of early-stage diagnoses dropped significantly. Conversely, the proportion of advanced-stage diagnoses increased markedly during the pandemic: stage IV diagnoses significantly increased from 13.1% in 2018 to 20% in 2021. These shifts indicate that many cases of prostate cancer were not detected until they had progressed to higher-risk Gleason classes and more advanced stages. Similarly, an Italian study by Ferrara et al. [17] reported a significant reduction in new cancer diagnoses during the lockdown, which likely contributed to the increase in advanced-stage presentations and subsequent shifts in therapy requirements. Accordingly, Nossiter in the UK showed that in 2020, compared to 2019, patients were diagnosed at more advanced stages (stage IV: 21.2% vs. 17.4%) and at slightly older ages (57.9% vs. 55.9% ≥ 70 years; *p* < 0.001) [30]. Pepe et al. also showed an observed increase in the proportion of cases presenting with advanced (pT3b: 11.2 vs. 25.6%; nodal-positive: 14.8 vs. 46.1%) and metastatic prostate cancer (5.9 vs. 9.3%), possibly due to an expected low incidence of clinical visits, prostate biopsy, and enrolment in active surveillance protocols during the COVID-19 pandemic [31]. Another critical point worth discussing is the therapeutic approach to prostate cancer during the COVID-19 pandemic. Our data showed that the rate of surgical interventions remained relatively stable, at around 35%, while radiotherapy use dropped slightly in 2020, with a subsequent rebound to 42% in the post-COVID-19 year, which indicates a recovery phase where efforts were made to address the backlog of cases that had accumulated during the peak pandemic period. Notably, the distribution of surgical interventions might have been influenced by the pandemic. There was a notable reduction in surgeries among patients aged 70–80 during 2020, likely reflecting a strategic decision to limit surgical risks for older patients more vulnerable to severe COVID-19 outcomes. This approach aligns with recommendations from various oncological societies to carefully consider the risk–benefit ratio of delaying surgery versus inducing potential COVID-19 exposure [32]. Accordingly, Nossiter et al. [30] described a reduction in both the number of radical prostatectomies (RPs) in 2020 by 26.9% and the number of treatments by external beam radiotherapy by 14.1%. Other changes included a reduced use of docetaxel chemotherapy in men with hormone-sensitive metastatic prostate cancer with a related increase in the use of enzalutamide [30]. Moreover, another Italian study showed that the frequency of open RPs increased compared with those using a laparoscopic approach. Furthermore, a higher percentage of men received external radiotherapy (25.1% vs. 54.2%) [31]. The decline in surgical interventions is notable, especially in light of a recent comprehensive literature review on the effectiveness of radical RP and radiation therapy versus observation in managing prostate cancer [33]. This review found that RP significantly improved overall survival both in observational studies and randomized controlled trials, establishing it as a strong option in prostate cancer treatment. This consistent benefit of RP suggests its pivotal role in prolonging survival among prostate cancer patients when compared to the passive approach of observation. A significant fact that emerges from our analysis is that the pandemic abruptly halted the increasing trend in prostate cancer observed in previous years, while the mortality rate remained stable. In our study, the incidence trend showed an age-adjusted standardized rate of 137.9 × 100,000 in 2010; this had reached a value of 149.5 × 100,000 by 2019, and the trend would probably have increased in subsequent years if it had not have been for the dramatic event of the pandemic, recording a TSD of 113.2 in 2020 and 105.9 in 2021. The increase in prostate cancer incidence observed until 2019 is consistent with trends in many Western countries, which can be largely attributed to the sometimes inappropriate use of PSA as a diagnostic tool.

The observed reductions in certain cases are partially attributable to the decline in PSA testing. Introduced in 1986, the PSA test has been a pivotal tool for the early diagnosis and management of prostate cancer, aiding in identifying men who may require a prostate biopsy, assessing therapeutic responses, monitoring tumor progression, and screening for prostate cancer [34,35,36]. Although PSA is not a definitive diagnostic test, as elevated levels can also occur in benign prostatic hyperplasia and other non-cancerous conditions [37,38,39], rapid increases in PSA values are often associated with prostate cancer. The widespread use of PSA testing has led to earlier diagnoses and improved survival rates due to early detection [40,41]. Consequently, prostate cancer remains a prevalent neoplasm in Western countries [42,43], with increasing incidence rates observed in younger populations in Italy, particularly under the age of 50, showing an APC of +3.4 [2]. An analysis of American SEER spanning 2004 to 2018 has indicated that there was an increase in metastatic prostate cancer in the 45–74 age group from 2010 to 2018 (APC, 5.3%; 95% CI, 4.5% to 6.0%; *p* < 0.001), and a more pronounced rise in diagnoses among individuals aged 75 and older from 2011 to 2018 (APC, 6.5%; 95% CI, 5.1% to 7.8%; *p* < 0.001) [44]. Despite its limitations, the PSA test remains a crucial component in the early detection and management of prostate cancer. Changes in U.S. screening practices have decreased the overall incidence, but it is unclear if relaxed screening has affected advanced and metastatic prostate cancer rates. From 2007 to 2013, the annual incidence of metastatic prostate cancer rose (APC 7.1%), especially in men aged 55–69, with a 92% increase from 2004 to 2013 [45]. In this respect, another American study indicates that among men younger than 75 years, the proportion presenting with distant metastases increased from 2.7% (95% CI, 2.5–2.9%) to 4.0% (95% CI, 3.8–4.2%). For men aged 75 years or older, the proportion with distant metastases rose from 6.6% (95% CI, 6.2–7.0%) to 12.0% (95% CI, 11.2–12.7%) [46]. This finding is corroborated by a population-based study that shows a significant increase in distant-stage prostate cancer from 2010 to 2017 [47]. Several factors likely contributed to these changes. In addition to the lockdown measures, including the “I stay at home” decree in Italy, which caused significant disruptions in routine healthcare services and cancer screening programs, patients may have been hesitant to seek medical attention due to a fear of contracting COVID-19. These factors further contributed to delays in diagnosis. The global COVID-19 pandemic resulted in a range of impacts, with particularly severe consequences for cancer patients [48,49]. Beyond the mortality associated with COVID-19 itself, the experience of cancer patients has been especially dire. This heightened vulnerability is attributable to two main factors, which are extensively documented in the literature over the past two years. Firstly, cancer patients exhibit increased susceptibility to COVID-19 infection, which consequently led to a higher likelihood of severe adverse events and mortality [50,51], particularly prior to the availability of vaccines. On the other hand, it is intriguing to note that our analysis indicates that although the pandemic interrupted the previously increasing trend in early detection of prostate cancer, the mortality rate remained stable. However, a longer follow-up period might be essential to better capture potential differences, providing a more comprehensive understanding of the pandemic’s impact on prostate cancer outcomes. This approach aligns with the existing literature that emphasizes the importance of long-term data in assessing changes in cancer mortality trends. In this respect, a recent Cochrane review [52] showed significant oncological benefits of RP compared to watchful waiting. However, these benefits were only realized in men with a life expectancy exceeding 10 years. Strengths of this study include that was used high-quality data, guaranteed by a high percentage of microscopic confirmations and an absence of DCO data. The availability of long incidence trends and comprehensive data on stage, Gleason score, surgery, and radiotherapy enhances the study’s reliability. However, its limitations include its lack of comorbidity information. More importantly, the study only includes one Cancer Registry, which restricts the generalizability of the results. Unfortunately, in Italy, data for 2016 and estimates for 2020 have been published, making it not possible to assess the impact of COVID-19 on new diagnoses using estimates: real data are needed. While the lack of comparison with other Cancer Registries represents a clear limitation of the study, it also reflects a strength linked to the operators’ efforts in promptly recording 2021 data. Future population-based studies incorporating data from multiple Cancer Registries could provide a more comprehensive understanding of the impact of COVID-19 on prostate cancer diagnoses and outcomes. Finally, it would be valuable to rapidly extend the assessment of COVID-19’s impact on new diagnoses, including 2022 data, to ascertain whether there has been a partial resumption of diagnostic and therapeutic activities.

## 5. Conclusions

The abrupt implementation of stringent lockdown measures led to the closure of or reductions in health activities and the suspension of cancer screening programs. This interruption in routine healthcare services significantly affected the timely diagnosis of prostate cancer, resulting in a notable decline in new cancer diagnoses during the pandemic period. As a consequence, our analysis revealed a concerning trend of increased metastatic forms and decreased early-stage presentations of prostate cancer during the pandemic. This shift towards diagnosing prostate cancer at more advanced stages underscores the critical importance of continuous cancer screening and early detection initiatives, even during global health crises. The observed increase in metastatic forms highlights the potential consequences of delayed diagnosis and underscores the urgent need for strategies to mitigate the impact of pandemic-related disruptions on cancer care. Despite the challenges posed by the pandemic, our study found no significant impact on the rates of surgical intervention and radiotherapy for prostate cancer patients. This suggests that, despite the disruptions in healthcare services, essential treatments for prostate cancer patients were largely maintained. However, it is essential to remain vigilant and monitor long-term outcomes in order to assess any potential delays or adverse effects on treatment delivery and patient outcomes. Finally, this work underscores our team’s ongoing effort to explore the impact of COVID-19 on various cancers [53,54,55,56,57], through the use of real-world data to assess diagnostic delays and screening adherence.

## Figures and Tables

**Figure 1 biology-13-00499-f001:**
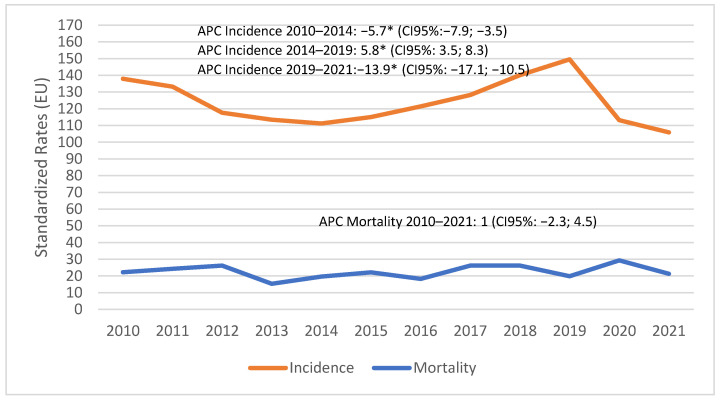
Reggio Emilia Cancer Registry. Prostate cancer 2010–2021. Incidence and mortality trends of prostate cancer. APC = annual percent change. * statistically significant.

**Table 1 biology-13-00499-t001:** Reggio Emilia Cancer Registry. Prostate cancer, years 2018–2021. Number of cases by age, Gleason score, stage, surgery, and radiotherapy.

	All (1123)	
	Mean	sd
Age at diagnosis	71.8	8.3
Age at diagnosis (groups)	n	%
≤70	472	42.1
71–80	489	43.5
80+	162	14.4
Gleason		
6	286	25.5
7	583	51.9
8–10	246	21.9
Unknown	8	0.7
Stage		
I	410	36.5
II	294	26.2
III	197	17.5
IV	169	15.1
Unknown	53	4.7
Surgery		
Yes	393	35.0
No	730	65.0
Radiotherapy		
Yes	429	38.2
No	694	61.8

**Table 2 biology-13-00499-t002:** Reggio Emilia Cancer Registry. Prostate cancer, years 2018–2021. Number of cases by age, Gleason score, stage, surgery and radiotherapy, and year of diagnosis.

	2018		2019		2020		2021		*p*-Value *
	Mean	sd	Mean	sd	Mean	sd	Mean	sd	
Age at diagnosis	71.6	7.9	72.5	8.1	70.9	8.1	71.9	9.0	
Age at diagnosis (groups)	n	%	n	%	n	%	n	%	
≤70	124	40.7	131	38.6	111	47.4	106	43.3	0.34
71–80	145	47.5	152	44.8	99	42.3	93	38.0	0.08
80+	36	11.8	56	16.5	24	10.3	46	18.8	0.16
Gleason									
6	79	25.9	96	28.3	66	28.2	45	18.4	0.11
7	162	53.1	171	50.4	116	49.6	134	54.7	0.87
8–10	62	20.3	71	20.9	50	21.4	63	25.7	0.20
Unknown	2	0.7	1	0.3	2	0.9	3	1.2	
Stage									
I	118	38.7	141	41.6	83	35.5	68	27.7	<0.05
II	73	23.9	74	21.8	72	30.8	75	30.6	<0.05
III	48	15.7	63	18.6	37	15.8	49	20.0	0.38
IV	40	13.1	44	13.0	36	15.4	49	20.0	<0.05
Unknown	26	8.5	17	5.0	6	2.5	4	1.6	
Surgery									
Yes	108	35.4	116	34.2	85	36.3	84	34.3	0.95
No	197	64.6	223	65.8	149	63.7	161	65.7	
Radiotherapy									
Yes	94	30.8	141	41.6	91	38.9	103	42.0	<0.05
No	211	69.2	198	58.4	143	61.1	142	58.0	
Total	305		339		234		245		

* Test for trend or chi^2^ as appropriate.

**Table 3 biology-13-00499-t003:** Reggio Emilia Cancer Registry. Prostate cancer, years 2018–2021. Number of cases by age and in patients undergoing surgery.

	2018		2019		2020		2021	
Age at diagnosis, group	n	%	n	%	n	%	n	%
≤70	72	66.7	66	56.9	52	61.2	59	70.2
71–80	35	32.4	47	40.5	31	36.5	23	27.4
80+	1	0.9	3	2.6	2	2.4	2	2.4
Stage								
I	26	24.1	21	18.1	22	25.9	18	21.4
II	43	39.8	35	30.2	38	44.7	29	34.5
III	35	32.4	52	44.8	18	21.2	31	36.9
IV	0	0.0	7	6.0	4	4.7	5	6.0
Unknown	4	3.7	1	0.9	3	3.5	1	1.2
Total	108		116		85		84	

## Data Availability

The data presented in this study are available on request from the corresponding author. The data are not publicly available due to ethical and privacy issues; requests for data must be approved by the Ethics Committee after the presentation of a study protocol.

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
