# Peer review of "Impact of the COVID-19 Pandemic on Prostate Cancer Diagnosis, Staging, and Treatment: A Population-Based Study in Northern Italy"

_biology, 2024, doi:10.3390/biology13070499_

Round 1

Reviewer 1 Report

Comments and Suggestions for Authors

In this paper the authors described a population based study in Northen Italy that demonstrated the huge impact of the COVID-19 Pandemic on prostate cancer diagnosis, staging and treatment, highlighting the need for ongoing diagnostic services and healthcare delivery even during global health emergencies. The results of the study are clearly presented and the discussion and conclusions were appropriately conducted.

However, some improvements are necessary:

1) The title should be include "in the Northern Italy", because of the different impact of COVID-19 pandemic between Northern and Southern Italy.

2) Line 117: please add "In the province of Reggio Emilia", between 2018 and 2021, a total of 1,123 cases of prostate cancer were registered."

3) Line 168: please, the authors indicate a reference for this statement: "province of Reggio Emilia, known for its high incidence of cancer and significant impact during the initial wave of the COVID-19 pandemic".

4) For clarity, the authors should specify that a Gleason score of 6 is low grade 7 is intermediate grade, and a score of 8 to10 is high grade cancer.

5) Fig. 1: please, move the text in the graph "APC incidence... and APC mortality.... " below the figure.

Comments on the Quality of English Language

 Minor editing of English language is required

Author Response

Comments and Suggestions for Authors

In this paper the authors described a population based study in Northen Italy that demonstrated the huge impact of the COVID-19 Pandemic on prostate cancer diagnosis, staging and treatment, highlighting the need for ongoing diagnostic services and healthcare delivery even during global health emergencies. The results of the study are clearly presented and the discussion and conclusions were appropriately conducted.

However, some improvements are necessary:

1) The title should be include "in the Northern Italy", because of the different impact of COVID-19 pandemic between Northern and Southern Italy.

RE: We appreciate the reviewer's insightful suggestion. Acknowledging the differential impact of the COVID-19 pandemic between Northern and Southern Italy, we have revised the title to specifically reference Northern Italy.

2) Line 117: please add "In the province of Reggio Emilia", between 2018 and 2021, a total of 1,123 cases of prostate cancer were registered."

RE: We thank the reviewer’s suggestion. We have added the specified clarification and included "In the province of Reggio Emilia" as recommended.

3) Line 168: please, the authors indicate a reference for this statement: "province of Reggio Emilia, known for its high incidence of cancer and significant impact during the initial wave of the COVID-19 pandemic".

RE: As requested, we have now added an appropriate reference to support the statement regarding the high incidence of cancer and the significant impact during the initial wave of the COVID-19 pandemic in the province of Reggio Emilia.

4) For clarity, the authors should specify that a Gleason score of 6 is low grade 7 is intermediate grade, and a score of 8 to10 is high grade cancer.

RE: We appreciate the reviewer's suggestion for clarity. We have specified that a Gleason score of 6 is low grade, 7 is intermediate grade, and a score of 8 to 10 is high grade cancer in the Materials and Methods section.

5) Fig. 1: please, move the text in the graph "APC incidence... and APC mortality.... " below the figure.

RE: As recommended by the reviewer, we have modified Figure 1 accordingly, moving the text "APC incidence... and APC mortality..." below the figure.

Comments on the Quality of English Language

 Minor editing of English language is required

RE: Thanks for the suggestion. We have conducted another thorough linguistic revision to improve the clarity and quality of the English language throughout the manuscript.

Reviewer 2 Report

Comments and Suggestions for Authors

This study provides valuable insights into the impact of the COVID-19 pandemic on prostate cancer management in Northern Italy. The manuscript is well-written, and its strengths lie in its comprehensive data analysis and its emphasis on the need for a robust healthcare system to manage prostate cancer. It is highly relevant as it demonstrates how the pandemic affected the diagnosis of prostate cancer patients and offers a detailed analysis of key aspects such as incidence rates, staging, Gleason scores, and treatment modalities. Additionally, this research contributes significantly to the existing literature by providing empirical data on the pandemic's impact on prostate cancer management, which can inform future healthcare policies and preparedness plans. However, the manuscript could be improved by expanding the scope to include a broader geographical perspective.

Comments:

1.       The study is restricted to a specific region in Northern Italy, which may limit the robustness and generalizability of the findings to other regions or countries with different healthcare systems and pandemic responses. The study could benefit from a comparative analysis with regions or countries that experienced different pandemic impacts or had different healthcare responses, providing a more comprehensive understanding of the variations in impact.

2.       The data only extends to one year post-pandemic (2021), which may not fully capture the long-term impact of COVID-19 on prostate cancer diagnosis and treatment. How was the outcome in 2022? This analysis can have crucial impact on the outcome of this study.

3.       This study focusses on prostate cancer, have author conducted some analysis on other cancer types to show broader impact?

4.       Does author encountered any discrepancies or gap in the data collection and reporting during the pandemic, if any, author may want to indicate that.

5.       Author may also want to add few sentences if about any changes in treatment protocols or innovations were occurred as a response to pandemic challenges.

Author Response

  1. The study is restricted to a specific region in Northern Italy, which may limit the robustness and generalizability of the findings to other regions or countries with different healthcare systems and pandemic responses. The study could benefit from a comparative analysis with regions or countries that experienced different pandemic impacts or had different healthcare responses, providing a more comprehensive understanding of the variations in impact.

RE: We thank the reviewer for the insightful suggestion. Unfortunately, in Italy, only data from 2016 and estimates for 2020 are available, making it infeasible to assess the impact of COVID-19 on new diagnoses using estimates instead of real data.

While this study's limitation is the lack of comparison with other cancer registries, it also highlights a strength: the dedicated effort of our team to promptly record the 2021 data. We have added this point to the discussion section.

  1. The data only extends to one year post-pandemic (2021), which may not fully capture the long-term impact of COVID-19 on prostate cancer diagnosis and treatment. How was the outcome in 2022? This analysis can have crucial impact on the outcome of this study.

RE: Thanks for the valuable suggestion. Unfortunately, the collection of incidence data requires several months of work. We acknowledge the importance of extending the analysis to include data from 2022 to fully capture the long-term impact of COVID-19 on prostate cancer diagnosis and treatment. We welcome the reviewer's proposal and plan to include 2022 data in future analyses to assess whether there has been a partial resumption of diagnostic and therapeutic activities. In this respect, we have added a sentence in the Discussion section.

  1. This study focuses on prostate cancer, have author conducted some analysis on other cancer types to show broader impact?

RE: Thanks for the request. Our team has conducted extensive research on the impact of COVID-19 on various cancer types. We have included references to some of these studies in the discussion to provide a broader context.

  1. Does author encountered any discrepancies or gap in the data collection and reporting during the pandemic, if any, author may want to indicate that.

RE: The reviewer's question is reasonable. We did not encounter any discrepancies or gaps in data collection and reporting during the pandemic. The availability of information sources remained consistent. However, the work faced some challenges as certain registrars were temporarily reassigned to COVID-19 patient surveillance activities.

  1. Author may also want to add few sentences if about any changes in treatment protocols or innovations were occurred as a response to pandemic challenges.

RE: Regarding your request, there have been notable changes during the pandemic, such as reduced availability of operating rooms, interruptions to oncological screenings, and redirection of clinical resources towards managing Covid emergencies. In our province, Covid-19 has had a profound impact, resulting in numerous hospitalizations and unfortunately, several deaths. These challenges have been addressed in the discussion section.

Reviewer 3 Report

Comments and Suggestions for Authors

The paper by Lee et al. proposes a CNN-based architecture for melanoma. It deploys an attention mechanism that emphasizes high-frequency features comprises a Convolutional Self-Attention Block that combines channel and spatial attention. The paper claims improvements in various high-frequency features, such as contours and textures, compared with other deep learning methods. The manuscript is well-written, well-presented, and easy to follows. My suggestions are as follows:

1) From the title, paper seems to address the problem of enhancing medical image quality. However, from the Introduction it seems to focus on Melanoma. The beginning of the Introduction should focus on the problem of improving medical image quality, rather than Melonoma.

2) Related work (state-of-the-art) is very shallow and needs enhancement to incorporate more recently published paper.

3) The proposed method has been compared with two very old methods published in 2016 & 2017. Hence, the comparison must be done with recently published work. 

4) Description and more elaboration of Figures within the text are required.

5) The limitations of the current work and future directions are missing. It must be present in the conclusion section.  

6) Cite the following paper in the Introduction section: https://doi.org/10.1016/j.eswa.2022.116968 

Author Response

The paper by Lee et al. proposes a CNN-based architecture for melanoma. It deploys an attention mechanism that emphasizes high-frequency features comprises a Convolutional Self-Attention Block that combines channel and spatial attention. The paper claims improvements in various high-frequency features, such as contours and textures, compared with other deep learning methods. The manuscript is well-written, well-presented, and easy to follows. My suggestions are as follows:

1) From the title, paper seems to address the problem of enhancing medical image quality. However, from the Introduction it seems to focus on Melanoma. The beginning of the Introduction should focus on the problem of improving medical image quality, rather than Melonoma.

2) Related work (state-of-the-art) is very shallow and needs enhancement to incorporate more recently published paper.

3) The proposed method has been compared with two very old methods published in 2016 & 2017. Hence, the comparison must be done with recently published work. 

4) Description and more elaboration of Figures within the text are required.

5) The limitations of the current work and future directions are missing. It must be present in the conclusion section.  

6) Cite the following paper in the Introduction section: https://doi.org/10.1016/j.eswa.2022.116968 

RE: This request is not relevant to our paper, as our study focuses on COVID-19 and prostate cancer, not melanoma. It appears there may have been a mistake by the editorial staff regarding the subject matter.

Round 2

Reviewer 3 Report

Comments and Suggestions for Authors

The Study by Mangone et al. performs analysis of Cancer Registry data (2018-2021) from a Northern Italian. The manuscript is very hard to follow in the current form. It needs major revision

1) Abstract is too tightly written, which is very hard to follow.

2) In fact, authors tried to analyze impact of COVID-19 pandemic on three different aspects of Prostate Cancer, viz.  diagnosis, staging, and treatment. It make the topic of research very broad.

3) Existing studies on a similar theme (Review of literature) is missing in the paper.

4) Materials and methods are very concise. In fact, this section specially describes about the datasets. Authors must mention list of analysis tools and techniques, along with enough description.

5) The results are also very thin, consisting of few tables and very few explanations. Some of the Tables do not convey any meaning. 

6) Discussion and conclusion are well written.

7) Cite the following paper in introduction section: https://doi.org/10.2174/2666796702666210114110013

Author Response

Comments and Suggestions for Authors

The Study by Mangone et al. performs analysis of Cancer Registry data (2018-2021) from a Northern Italian. The manuscript is very hard to follow in the current form. It needs major revision

1) Abstract is too tightly written, which is very hard to follow.

RE: Thanks to the reviewer for the suggestion. We tried to rewrite the abstract hoping that it would now make it easier to read.

2) In fact, authors tried to analyze impact of COVID-19 pandemic on three different aspects of Prostate Cancer, viz.  diagnosis, staging, and treatment. It makes the topic of research very broad.

RE: Thanks to the reviewer for the comment. In fact, an objective of this type appears too discursive. We have better reformulated the aim of the study, focusing on the impact of Covid-19 on the shift towards a more advanced stage and, secondly, on treatments.

3) Existing studies on a similar theme (Review of literature) is missing in the paper.

RE: Thanks to the reviewer for the suggestion. We have added other references in the introduction.

4) Materials and methods are very concise. In fact, this section specially describes about the datasets. Authors must mention list of analysis tools and techniques, along with enough description.

RE: In fact the materials and methods section is too concise. We have integrated the section by better specifying the analyzes that were carried out.

5) The results are also very thin, consisting of few tables and very few explanations. Some of the Tables do not convey any meaning. 

RE: Thanks for the suggestion. We have better integrated the results section by integrating and better explaining the meaning of the results reported in the table.

6) Discussion and conclusion are well written.

RE: Thanks for this comment.

7) Cite the following paper in introduction section: https://doi.org/10.2174/2666796702666210114110013

RE:Thanks, we added this reference in the introduction, as suggested by the reviewer.